# Molecular Mode of Action of TRAIL Receptor Agonists—Common Principles and Their Translational Exploitation

**DOI:** 10.3390/cancers11070954

**Published:** 2019-07-07

**Authors:** Harald Wajant

**Affiliations:** Division of Molecular Internal Medicine, Department of Internal Medicine II, University Hospital Würzburg, 97080 Würzburg, Germany; harald.wajant@mail.uni-wuerzburg.de; Tel.: +49-931-201-71000

**Keywords:** antibody, antibody fusion proteins, apoptosis, cancer therapy, cell death, death receptors, TNF superfamily, TNF receptor superfamily, TRAIL

## Abstract

Tumor necrosis factor-related apoptosis-inducing ligand (TRAIL) and its death receptors TRAILR1/death receptor 4 (DR4) and TRAILR2/DR5 trigger cell death in many cancer cells but rarely exert cytotoxic activity on non-transformed cells. Against this background, a variety of recombinant TRAIL variants and anti-TRAIL death receptor antibodies have been developed and tested in preclinical and clinical studies. Despite promising results from mice tumor models, TRAIL death receptor targeting has failed so far in clinical studies to show satisfying anti-tumor efficacy. These disappointing results can largely be explained by two issues: First, tumor cells can acquire TRAIL resistance by several mechanisms defining a need for combination therapies with appropriate sensitizing drugs. Second, there is now growing preclinical evidence that soluble TRAIL variants but also bivalent anti-TRAIL death receptor antibodies typically require oligomerization or plasma membrane anchoring to achieve maximum activity. This review discusses the need for oligomerization and plasma membrane attachment for the activity of TRAIL death receptor agonists in view of what is known about the molecular mechanisms of how TRAIL death receptors trigger intracellular cell death signaling. In particular, it will be highlighted which consequences this has for the development of next generation TRAIL death receptor agonists and their potential clinical application.

## 1. Introduction

Tumor necrosis factor-related apoptosis-inducing ligand, also called apoptosis-2 ligand /Apo2L) has been cloned due to its homology to the apoptosis-inducing death ligands tumor necrosis factor (TNF) and cluster of differentiation 95 CD95) ligand (CD95L)/FasL. TNF, but particularly CD95L and TRAIL, induce apoptosis in a variety of transformed cell lines by stimulation of a group of related receptors, the so-called death receptors (DRs). Early on, there were attempts in preclinical models to exploit the apoptotic activity of TNF receptor-1 (TNFR1) and CD95, the DRs of TNF and CD95L, for cancer treatment. Rapidly, it turns out that the systemic activation of TNFR1 and CD95 results in life-threatening side effects. In the case of TNFR1, due to its strong proinflammatory effects which typically override its apoptotic activity and in the case of CD95 due to apoptosis induction in hepatocytes. In contrast, systemic TRAIL death receptor activation proved to be well tolerable. TRAIL and TRAIL death receptor agonists attracted therefore an enormous interest as possible anti-cancer drugs. Clinical trials with poorly active TRAIL agonists revealed good safety profiles but also showed only limited anti-cancer effects, perhaps not surprising in view of their submaximal activity. The improved understanding of the molecular mechanisms of TRAIL death receptor activation in the recent decade led to the rational development of a variety of novel TRAIL death receptor agonists with highly-specific activity. Although there is initial evidence that highly-active TRAIL death receptor agonists are hepatotoxic, some of them promise tumor-localized activity. Thus, the novel highly-active TRAIL death receptor agonists may fulfill the so far disappointed hope set on TRAIL death receptors agonists as cancer therapeutics.

## 2. Death Signaling by Tumor Necrosis Factor-Related Apoptosis-Inducing Ligand Receptors 1 and 2

### 2.1. Tumor Necrosis Factor-Related Apoptosis-Inducing Ligand and Its Receptors

Tumor necrosis factor-related apoptosis-inducing ligand is in most aspects a typical member of the TNF superfamily (TNFSF). It is a type II transmembrane protein of 281 amino acids in which the characteristic C-terminal TNF homology domain (THD) is linked to the transmembrane domain and the intracellular domain by a stalk region of app. 70 aa (Figure 1). The THD is responsible for ligand trimerization and receptor binding [1]. The stalk region stabilizes the trimeric structure of the molecule and contains cleavage sites for proteolytic release of soluble TRAIL [2,3]. The release of soluble TRAIL was early on reported after its cloning for various T- and B-cell lymphoma cell lines and later then also for neutrophils and γδ T-cells [4,5,6]. Soluble TRAIL levels in serum have been furthermore evaluated in a variety of studies for various diseases (Table 1). However, these studies only reported association/correlation with serum TRAIL levels. Thus, it is unclear whether TRAIL is of functional relevance in these diseases.

The regulation of TRAIL processing is poorly investigated and the proteases involved have not been conclusively identified although there is initial evidence for a role of ADAM33 and cathepsin E (CTSE) [18,19]. Unique for a ligand of the TNFSF, membrane-bound as well as soluble TRAIL trimers contain a zinc ion coordinated by an unpaired cysteine residue in the THD (Cys-230). Zinc chelation of TRAIL is crucial for the assembly and maintenance of functionally-folded ligand trimers [20,21,22]. TRAIL is commonly expressed by various immune cells including T-cells, neutrophils, dendritic cells, plasmacytoid dendritic cells, innate lymphocytes, and NK- and NK T-cells upon activation by type I and type II interferons [23,24,25,26,27,28,29,30]. In accordance with its immune cell-associated expression pattern and its apoptosis-inducing activity, various studies demonstrated a role of TRAIL in tumor surveillance [31]. TRAIL expression has also been reported for vascular smooth muscle cells, keratinocytes, intestinal epithelial cells, and mammary epithelial cells [32,33,34,35]. The functional/biological relevance of TRAIL expression by these cell types, however, is largely unknown.

Typical for a ligand of the TNFSF, TRAIL interacts with receptors of the TNF receptor superfamily (TNFRSF). The receptors of the TNFRSF are characterized by having one to six copies of a cysteine-rich domain (CRD) in their extracellular part [31,36]. There are five different types of TNFRSF receptors that bind TRAIL with high affinity (Figure 2). TRAIL receptor-1 (TRAILR1, DR4) and TRAILR2 (DR5) belong to the DR subgroup of the TNFRSF. As such, TRAILR1 and TRAILR2 have a protein–protein interaction domain, called death domain (DD), in their cytoplasmic part which links these receptors to cytotoxic signaling cascades. TRAILR3 (DcR1) is a glycosylphosphatidylinositol (GPI)-anchored receptor without its own intrinsic signaling abilities. TRAILR4 contains a truncated seemingly non-functional DD (tDD) and has been identified as an antagonist of TRAIL-induced cell death [36,37]. There is some evidence from ectopic expression experiments that TRAILR4 can trigger NFκB and PI3K/Akt signaling but these possibilities have not been studied yet in more detail [38,39]. Osteoprotegerin (OPG) is a soluble decoy receptor for TRAIL which also binds RANKL, another ligand of the TNFSF, which, via its receptor RANK, regulates osteoclastogenesis but also differentiation and activation of T-cells [40].

The two TRAIL death receptors share approximately 70% sequence homology and form together with CD95, a distinct category of DRs. The DRs of this category are distinguished from other DRs by the fact that activation of the apoptosis-inducing procaspase-8 molecule occurs directly in the plasma membrane-associated ligand- and receptor-containing death-inducing signaling complex (DISC) and not in a ligand-induced secondarily formed cytoplasmic-signaling complex [43]. As for some other receptors of the TNFRSF, there is evidence that TRAILR1, TRAILR2 and TRAILR4 undergo low affinity (KD > 1 µM) homotypic and heterotypic dimerization in the absence of TRAIL which facilitates ligand binding [44,45,46]. The ability of TRAILR4 to heteromerize with TRAILR1 and TRAILR2 suggests that TRAILR4 not only antagonizes TRAIL death receptors by competition for ligand binding but also by forming heteromeric complexes with TRAILR1/2 with reduced activity [45,46,47]. Noteworthy, there is evidence that liganded homotypic TRAILR1- and TRAILR2-complexes have superior activity compared to liganded heterotypic TRAILR1/2 complexes [48,49]. All TRAIL receptors are glycosylated and a few studies investigated the relevance of glycosylation for the cytotoxic activity of TRAILR1 and TRAILR2. It has been reported that N-glycosylation of the murine TRAIL death receptor and human TRAILR1 as well as O-glycosylation of human TRAILR2 promote apoptotic signaling [50,51]. A third study, however, reported enhanced TRAIL binding and cell death-induction when N-glycosylation of the mouse TRAIL death receptor has been prevented [52].

### 2.2. TRAIL Death Receptor-Induced Engagement of Cytotoxic Signaling Pathways

As briefly addressed above, TRAIL receptors interact in unstimulated cells homo- and heterotypically with low affinity (Figure 2 and Figure 3). The corresponding interaction side of the TRAIL receptors, designated as pre-ligand-binding assembly domain (PLAD), resides in the N-terminal first CRD1 of the receptors and physically separates from the TRAIL-binding side formed by CDRs 2 and 3. PLAD-promoted receptor dimerization and trimerization have been discussed. Noteworthy, there is recent evidence that, at least in the case of TRAILR2, the transmembrane helix possesses distinct areas mediating trimerization and dimerization [53]. In particular, the latter has been implicated by mutagenesis in unliganded TRAILR2 auto-association and thus may cooperate with the PLAD in forming TRAIL-free receptor dimers [53]. There is, furthermore, evidence that the TRAILR2 ectodomain hinders the formation of oligomeric networks via the cooperate action of the dimerization and trimerization areas of the TM domain [53]. In view of the weak auto-affinity of TRAIL receptors, it is currently unclear which fraction of the TRAIL receptors are actually in the dimeric/trimeric state in the absence of TRAIL. It is, however, tempting to speculate that TRAIL preferentially interacts with the pre-assembled receptor species. Crystallographic studies clearly show that a TRAIL trimer interacts with three TRAIL receptor molecules which bind to the interfaces formed of the symmetrically assembled three protomers of a TRAIL trimer [54,55,56]. It has been furthermore observed that apoptosis-inducing liganded TRAILR2 complexes form oligomeric TRAILR2 networks under crucial involvement of dimeric receptor–receptor interactions [53,57,58] The TRAILR2 TM domain has also been implicated in the dimeric interaction of the liganded TRAILR2 molecules while the relevance of the PLAD has not been evaluated in this context, yet [53]. Already in one of the two studies reporting cloning of human TRAIL (and frequently afterwards), it has been shown that soluble TRAIL has a relative low capacity to trigger apoptosis as long as it is not secondarily oligomerized (Figure 3). This suggests that trimeric soluble TRAIL-TRAILR complexes fail to interact secondarily and are thus poorly active/apoptotic. With TRAIL in its membrane-bound form, however, secondary clustering of trimeric TRAIL-TRAILR complexes to receptor networks with high cell death-inducing activity may occur spontaneously due to the concerted action of the supra-high molecule concentrations (µM-mM) in the cell-to-cell contact zone and the low intrinsic auto-affinity of TRAILRs. In the context of the supramolecular TRAIL-TRAIL death receptor complexes, the cytosolic DD-containing adapter protein Fas-associated death domain protein (FADD) interacts with its DD with the DD of TRAILR1/2. Besides its DD, FADD contains a second DD-related protein–protein interaction domain called death effector domain (DED). By virtue of its DED, TRAILR1/2-bound FADD interacts with one of the two DEDs contained in procaspase-8 and recruits this apoptosis-inducing key molecule [46]. Thus, a single procaspase-8 molecule interacts in an asymmetrical manner with its two DEDs with two death receptor-bound FADD adapter proteins. Death receptor–FADD complex-bound procaspase-8 is able to directly recruit additional procaspase-8 molecules by DED–DED interactions resulting in procaspase-8 chains [59,60,61]. Now, it has been furthermore shown in cell-free experiments that the procaspase-8 prodomain forms filaments composed of three parallel helical prodomain chains that nucleate on FADD oligomers [59]. It is therefore tempting to speculate that six death receptor-bound FADD molecules, and thus two death receptor trimers, are required to form a platform stimulating the assembly of procaspase-8 filaments (Figure 3). In the death receptor-associated procaspase-8 filaments/chains, neighboring procaspase-8 molecules dimerize and undergo autocatalytic two-step processing. This results in mature heterotetrameric caspase-8 molecules which are released in the cytoplasm to trigger the execution steps of apoptosis [31,37]. In sum, it appears that the need for clustering of trimeric TRAIL-TRAIL death receptor complexes for efficient apoptosis induction reflects the special mechanisms of caspase-8 activation.

The requirement of a hexameric TRAIL-TRAILR1/2-FADD complex as a condensation nucleus for the formation of caspase-8-activating procaspase-8 filaments can also straightforwardly explain why bivalent antibodies against TRAIL death receptors are typically poor agonists as long as they are not oligomerized or bound to Fc*γ*-receptors (Fc*γ*Rs). Thus, protein G or anti-IgG oligomerized anti-TRAILR1/2 antibodies might bring together six or more TRAIL death receptors similar to crosslinked soluble TRAIL trimers (Figure 4). Secondary clustering of dimeric antibody-TRAILR1/2 complexes to receptor networks with high activity may spontaneously occur in the case of FcγR-bound antibodies in a similar fashion as with membrane TRAIL due to the high molecule concentrations in the contact zone between FcγR^+^ and TRAILR1/2 expressing cells and the weak auto-affinity of the TRAIL death receptors (Figure 4).

The idea that soluble TRAIL and bivalent anti-TRAILR1/2 antibodies are suboptimal triggers of apoptotic TRAILR1/2 signaling due to their poor ability to promote secondary clustering of liganded receptor trimers/dimers might furthermore explain the synergistic action of the non-competitive anti-TRAILR2 antibody Conatumumab (AMG655) and soluble TRAIL reported in some studies [62,63]. It is straightforwardly imaginable that trimeric soluble TRAIL-TRAILR2 complexes are secondarily oligomerized by the interaction with non-competing anti-TRAIL2 antibodies.

### 2.3. TRAIL-Stimulated Non-Cell Death Signaling

Although TRAIL and the TRAIL death receptors have mainly been studied with respect to apoptosis induction, over the years it became evident that TRAILR1 and TRAILR2 also engaged in various non-apoptotic signaling pathways including the strongly proinflammatory classical NFκB pathway [31,43]. FADD and caspase-8 are also required for TRAILR1/2-mediated NFκB signaling and occur via a complex in the cytoplasm secondarily formed without the need of the enzymatic activity of caspase-8. Similar to apoptosis induction, NFκB activation by soluble TRAIL is strongly enhanced by its crosslinking [43]. Thus, the TRAILR1/2-associated mechanisms discussed above for TRAIL-induced apoptosis might also be of relevance for initiating TRAIL death receptor-induced NFκB signaling. The TRAIL death receptors also trigger signaling pathways leading to the activation of MAP kinases and the PI3K/Akt pathway to stimulate proliferation and cell migration [31,43]. The molecular mechanisms by which TRAILR1 and TRAILR2 stimulate these events are largely unknown and may substantially differ from FADD/caspase-8-dependent NFκB signaling and apoptosis. It cannot be ruled out therefore that in these cases secondary clustering of initially formed trimeric TRAIL-TRAILR1/2 complexes is less important. Indeed, for death receptor CD95, which engages in apoptosis in a similar fashion as the TRAIL death receptors, FADD-independent activation of tyrosine kinases by soluble ligand trimers, has been demonstrated [43].

With respect to cancer therapy, one aspect of non-apoptotic TRAIL signaling appears particularly relevant. When initially TRAIL-sensitive tumor cells become resistant by mechanisms leaving non-apoptotic signaling intact, especially NFκB signaling, TRAIL and TRAIL death receptors can develop protumoral activity. TRAIL death receptors then become detrimental and TRAIL inhibitors might achieve therapeutic relevance. Indeed, protumoral activity of TRAIL has been reported in the last decade in several preclinical tumor models [64,65,66,67].

## 3. Conventional TRAIL Death Receptor Agonists and Their Limitations: The Lessons from Preclinical Studies and Clinical Trials

Four phase I to phase II clinical trials with recombinant soluble TRAIL (AMG 951, Dulanermin) have been completed, and one phase III trial is still active. All trials showed good and acceptable tolerance for dulanermin [31,37]. However, despite some clinical efficacy (reduced progression-free survival but no change in overall survival) in the ongoing phase III trial for the treatment of advanced non-small-cell lung cancer with a combination of Dulanermin, vinorelbine and cisplatin, the results were rather disappointing [31,37,68]. This is less surprising when one considers that Dulanermin, due to its trimeric nature, probably only engages a suboptimal, therapeutically-insufficient TRAIL death receptor response. Indeed, a Flag-tagged version of Dulanermin showed much stronger apoptotic activity on a variety of cancer cell lines in vitro after anti-Flag oligomerization and attachment to cell or liposomes also strongly enhanced it activity [69]. Safety and good tolerance have also been reported for a very poorly described stabilized TRAIL variant designated as circularly permuted TRAIL, CPT [70]. However, despite its improved in-vitro activity compared to conventional TRAIL, it is not expected that this soluble trimeric TRAIL variant unleashes maximal TRAIL death receptor activity in vivo.

Not at least due to the rapid elimination of soluble TRAIL from the serum (half-life of 23–31 min in nonhuman primates [71]), several groups and companies developed agonistic antibodies against TRAILR1 and TRAILR2 to overcome this limitation (for a non-exhaustive list see Table 2). A couple of them have been extensively tested in clinical trials [31,37]. As in the case of Dulanermin, excellent tolerance was recognized in these trials but again anti-tumoral efficacy was very limited [31,37]. In view of the increasing evidence that cytotoxic TRAIL death receptor-signaling is engaged by the co-action of six receptor molecules that form the assembly platform for the DISC, it appears less plausible that bivalent monospecific anti-TRAILR1 or anti-TRAILR2 alone can trigger a strong TRAIL death receptor response. The potential death receptor-stimulating effect of anti-TRAILR1/2 antibodies in the tumor is therefore probably crucially determined by FcγR-anchoring-dependent activity. The anti-TRAILR1/2 antibodies evaluated so far in clinical trials were not optimized for FcγR-binding and cannot overcome the limitations that result from the FcγR dependency of their agonism such as insufficient numbers of FcγR-expressing immune cells in the microenvironment, poor expression levels of FcγRs at a per cell level, competition with endogenous IgG molecules for FcγR binding, and unwanted effects due to FcγR activation.

Interestingly, very few groups reported generation of anti-TRAILR1/2 antibodies and scFvs which are highly active without oligomerization or FcγR binding (e.g., Refs. [74,81,82,86,87,88,89,90]). In most cases, the high oligomerization-independent agonistic activity of anti-TRAILR1/2 antibodies/scFvs has been poorly investigated and, therefore, a role of aggregated molecule species present in the antibody/scFv preparations cannot be fully ruled out. However, especially in the case of the anti-TRAILR2 KMTR2m various control experiments and analysis have been performed and gave no evidence for this possibility [77,91]. Future studies must show in detail whether and how such antibodies and scFvs trigger receptor oligomerization despite their only bivalent or even only monomeric nature.

In fact, the initial rationale to target TRAILR1 and TRAILR2 for cancer therapy was to trigger apoptotic cell death by engagement of these receptors. Preclinical studies with conventional anti-TRAIL death receptor antibodies also revealed antibody-dependent cell-mediated cytotoxicity (ADCC) and complementary dependent cytolysis (CDC) as a possible mode of action [92]. Thus, especially TRAILR2/DR5 with its p53/stress-induced expression might be exploited as a tumor marker [93].

Taken together, the less exciting antitumor activity of soluble TRAIL (Dulanermin, CPT) and conventional anti-TRAILR1/2 antibodies observed so far in clinical trials, may rather reflect the limited agonistic potential of these molecules rather than the lack of therapeutic efficacy of activated TRAIL death receptors. Whether this conclusion is correct must be evaluated in future clinical studies with TRAIL and TRALR1/2 variants of the next generation which unleash the full cytotoxic activity of TRAIL death receptors. It will be important to see whether the good tolerability of systemically applied TRAIL death receptor-targeting biologicals will be maintained when agonists with higher activity are considered. So far, there is indeed initial evidence that highly active TRAIL death receptor agonists could cause liver toxicity. First, a clinical study with a TRAILR2-specific tetrameric nanobody was terminated due to hepatotoxicity in some patients [94]. Second, it was found that repeated injections of human TRAIL into cynomolgus monkeys result in the generation of anti-TRAIL antibodies, TRAIL crosslinking and hepatotoxicity [95]. On the other side, the anti-TRAILR2 mAb Apomab, which is highly agonistic after crosslinking, showed no cytotoxic effects on human hepatocytes in vitro even if crosslinked [79].

## 4. Next Generation Ligand-Based TRAIL Death Receptor Agonists

### 4.1. Stabilized TRAIL Trimers and TRAILR1- and TRAILR2-Specific TRAIL Mutants

Soluble variants of TRAIL containing the THD are rather unstable and tend to form misfolded high molecular weight aggregates when expressed in eukaryotic cells [2]. It has been demonstrated that TRAIL contains an unpaired cysteine (Cys-230) located at a conserved position where CD95L and TNF have a disulfide-bridged cysteine [20]. The three Cys-230 of a TRAIL trimer coordinate a zinc ion, and this improves and stabilizes the trimeric assembly of the molecule. Interestingly, efficient zinc coordination of recombinant soluble TRAIL trimers takes place in bacterial expression systems, while in higher cells, ectopically-expressed soluble TRAIL majorly forms inactive disulfide-linked dimers under crucial involvement of Cys-230 [20]. There is, furthermore, evidence that the stalk region of TRAIL contributes to the formation of stable TRAIL trimers [2]. Several groups found that the physical, typically covalent linkage of three soluble TRAIL protomers results in TRAIL preparations with enhanced apoptotic activity compared to conventional soluble TRAIL preparations (Table 3). In most cases, the linkage of the three TRAIL protomers has been achieved by genetic fusion of a TRAIL protomer with a heterologous trimerizing domain, e.g., from tenascin-C or the surfactant protein-D [2,96]. Stabilizing linkage, however, is also possible by encoding the three soluble TRAIL protomers by a single cDNA expression cassette resulting in peptide-linker connected TRAIL trimers (scTRAIL) [97]. The physically-linked soluble TRAIL trimers, however, typically still require oligomerization to achieve maximal activity [2]. This suggests that stabilization of soluble TRAIL trimers does not fundamentally change the intrinsic receptor stimulatory capacity of soluble TRAIL trimers but rather increases and maintains this active molecule species in a TRAIL preparation. The question of how physical linkage of three protomers exactly stabilizes the active assembly of soluble TRAIL is currently not known, neither is its effect on zinc coordination or Cys-230 disulfide bridge formation; in addition, its impact on dissociation and/or reassociation of TRAIL trimers have been addressed.

It is worth mentioning that stabilized soluble TRAIL trimers can also be used as building blocks for the development of TRAIL fusion proteins which overcome the limitations of soluble TRAIL trimers such as poor serum half-life or limited specific activity. For example, a fusion protein of TNC-TRAIL with serum albumin showed improved serum retention and enhanced anti-tumor activity in nude mice in comparison to TNC-TRAIL [102]. TNC-TRAIL and scTRAIL have furthermore also been exploited as building blocks to construct soluble TRAIL fusion proteins displaying improved specific activity due to oligomeric assembly or cell surface anchoring (for details see below, Section 4.2).

### 4.2. TRAIL Variants with Superior Specific Activity

#### 4.2.1. TRAIL Fusion Proteins Containing Two or More TRAIL Trimers

The much higher activity of anti-Flag antibody oligomerized Flag-TRAIL and His-TRAIL, which tends to aggregate, early on suggested that soluble TRAIL trimers may possibly not allow the exploitation of the full apoptotic capacity of TRAIL death receptors. Since antibody crosslinked TRAIL and His-TRAIL preparations are rather non-defined mixtures of oligomers, their translational potential is limited. Based on the assumption that the close proximity of two or more liganded TRAILR1/2 trimers is the key step to ensure maximal apoptosis activation, various fusion proteins of TRAIL, which contain more than on TRAIL trimer, have been developed in recent years (Figure 5).

Genetic fusion of the TRAIL-related death ligand FasL with the dimeric Fc domain of human IgG1 resulted in a hexameric protein with an approximate 1000-fold higher apoptotic activity than Flag-FasL [103]. Although fusion with an Fc domain also strongly enhances the activity of other soluble TNFSF ligands, this approach failed largely for TRAIL [104]. Surprisingly, however, when two scTRAIL domains were dimerized by a Fc domain, a highly active molecule was formed [105]. Similarly, fusion proteins of scTRAIL with other oligomerization domains also display high apoptotic activity (Table 4). It is thus tempting to speculate that the stabilization of the trimeric assembly of TRAIL is of particular importance in the context of oligomeric TRAIL fusion proteins. It is worth mentioning that a recent publication reported high activity for an IgG1 Fc fusion protein of conventional TRAIL [106]. The striking difference to the previous study by Bossen et al., which reported a largely non-functional Fc-TRAIL, seems to be the production method. While Wang et al. produced Fc-TRAIL in bacteria (*E. coli*), Bossen et al. used a eukaryotic expression system [104,106]. This mirrors the situation discussed above for conventional soluble TRAIL. It is thus temping to speculate that eukaryotic production of TRAIL fusion proteins can overcome the need for stabilization of the trimeric TRAIL domain. Indeed, Adi-TRAIL, another recently reported highly-active hexameric TRAIL fusion protein, was also produced in *E. coli* [107].

#### 4.2.2. Cell Surface Anchoring TRAIL Fusion Proteins

The fact that the enforced proximity of two or more soluble TRAIL trimers alone is already fully sufficient to trigger strong activation of TRAIL death receptors, suggests that the superior activity of membrane TRAIL does not require special, primary sequence-encoded information. This opens the possibility, as discussed in detail above in Section 2.2, that it is the sole “capturing” to the plasma membrane that makes full-length TRAIL trimers potent activators of TRAIL receptor signaling. In accordance with this hypothesis, we and others showed that soluble TRAIL trimers genetically fused with an anchoring domain, which enables binding to a cell surface exposed structure, acquires membrane TRAIL-like high apoptotic activity (Table 5). Typically, cell surface antigen-specific scFvs have been used as anchoring domains but other protein domains recognizing molecular targets displayed on the cell surface have also been successfully used (Table 5).

It is worth mentioning that since the activity of such anchor domain-TRAIL fusion proteins (Figure 6A) is dependent on anchoring to the antigen/target recognized by the anchor domain, this type of TRAIL fusion protein is principally suited to construct TRAILR1/2 agonists with locally-restricted in-vivo activity. Thus, anchor domain-TRAIL fusion proteins can be essentially considered as TRAIL prodrugs “activated” by anchoring to cell surface displayed target structures (Figure 6B). This aspect could gain particular relevance in view of recent findings suggesting that highly-active TRAILR2 agonists are hepatotoxic [94,95] and/or in situations where systemic activation of TRAIL death receptors is accompanied by dose-limiting side effects, e.g., when combined with drugs sensitizing normal cells for TRAIL-induced apoptosis. In accordance with the idea that the sole cell surface anchoring is the decisive factor that converts soluble TRAIL trimers into molecules with a high TRAIL receptor-stimulating capacity, not only scFvs can been used to construct TRAIL fusion proteins with anchoring-dependent activity but also various other types of protein domains fulfilling two basic requirements (Table 5). First, the protein domain fused to TRAIL to allow cell anchoring must not interfere with trimeric TRAIL self-assembly, and secondly, should not lead to auto-aggregation and thus anchoring-independent agonistic activity.

For example, soluble TRAIL which has been equipped with the extracellular domain of the TNFRSF receptor CD40 revealed strongly enhanced TRAIL-induced apoptosis if anchored to the membrane-bound form of the TNFSF ligand CD40L [131]. Noteworthy, a corresponding TRAIL fusion protein containing the extracellular domain of Fn14 similarly displays enhanced activity in the presence of the soluble form of its ligand TWEAK (TNF-like weak inducer of apoptosis) [138]. Thus, the mode of action in this special case seems to be rather soluble factor-assisted oligomerization of soluble TRAIL trimers than cell anchoring-mediated conversion into a membrane TRAIL-like molecule. Indeed, membrane TWEAK contains a furin site in its stalk region and is thus efficiently processed to soluble TWEAK [139]. TRAIL fusion proteins intrinsically have the capacity to be bifunctional, in particular if the anchoring domain modifies the activity of the molecule recognized by the anchoring domain. Thus, the choice of an appropriate anchoring domain can allow the construction of TRAIL fusion proteins which, upon anchoring, stimulate apoptotic TRAIL signaling and concomitantly trigger effects which potentiate the effect of TRAIL death receptor activation. For example, T47D breast cancer cells which are TRAIL resistant due to autocrine CD40L-CD40 signaling can be efficiently killed by a fusion protein of soluble TRAIL with the extracellular domain of CD40 which links the anchoring of soluble TRAIL trimers with the blockade of the protective autocrine CD40L-CD40 survival loop [131]. Cell-anchored fusion proteins of soluble TRAIL may act in an autocrine fashion on cells expressing the anchor structure, but also stimulate TRAIL receptors in a paracrine fashion, therefore, having the ability to trigger cell death also in neighboring, potentially antigen-negative tumor cells [118,140].

The superior activity of membrane TRAIL might also be exploited therapeutically by inducing endogenous TRAIL expression. Indeed, the antitumoral activity of the small molecule ONC201 (TIC10) imipridone seems to be at least partly related to the upregulation of TRAIL expression [37,141].

## 5. Next Generation TRAIL Death Receptor Agonists Based on Antibodies and Related Molecules

The strategies to enhance the specific activity of anti-TRAIL death receptor antibodies are the same as in the case of soluble TRAIL trimers, namely increasing the number of TRAIL death receptor binding sites and cell surface anchoring. Indeed, oligomerization of antibody-bound TRAILR1/2 dimers, similarly to oligomerized TRAIL trimers, should eventually result in hexameric (or even more aggregated) receptor complexes allowing the engagement of intracellular signaling. Likewise, FcγR-bound dimeric antibody-TRAILR1/2 complexes may undergo spontaneous clustering due to the intrinsic weak TRAILR1/2 auto-affinity and the special conditions in the cell-to-cell contact zone.

### 5.1. Anti-TRAIL Death Receptor Variants with Increased Valency

Lee et al. investigated the effects of valency on agonistic activity on the example of a humanized scFv derived from the anti-TRAILR1 antibody AY4 [142]. Lee et al. achieved dimerization and trimerization of the scFv domain by linking it with a flexible linker to the N-terminus of a dimerizing leucine zipper and a trimerizing isoleucine zipper domain with His tag. While the monomeric scFv:TRAILR1 version showed no apoptotic activity at all, the dimerized and even more the trimerized variant displayed significant cytotoxic activity. Unfortunately, the effect of further crosslinking was not investigated. Thus, it became unclear in this study whether the trimerized scFv:TRAILR1 already unleashed the maximal apoptotic activity of TRAILR1 or whether it simply has a gradual increased agonisitic activity. In another study, the agonism of tri- tetra- and pentameric versions of a llama-derived TRAILR2-specific nanobody were compared with those of soluble TRAIL and the anti-TRAILR2 LBY135 [143]. While the trimeric nanobody variant turned out to be comparably active as soluble TRAIL and LBY135, the tetrameric nanobody showed >10 and a pentameric variant >100 fold increase in activity. The much higher specific activity of the tetrameric and pentameric nanobodies furthermore correlated with a more efficient recruitment of FADD and caspase-8 to TRAILR2 [143]. A first phase I study with TAS266, a tetrameric TRAILR2 nanobody variant, however, had to be stopped due to dose-limiting liver toxicity [94]. It is worth mentioning that this side effect correlated with the presence of preexisting antibodies in the patients recognizing TAS266 which might have enhanced the already highly-specific activity of this tetravalent TRAILR2 agonist.

A tetravalent antibody variant composed of four scFv domains derived from a non-agonistic TRAILR2-specific antibody was also found to be highly active [144]. Oligomeric assembly of the scFv:TRAILR2 domains was achieved in this case by genetic fusion with the p53 tetramerization domain and aa residues 490–513 of HSA. Since the p53 tetramerization domain is composed of two pairs of anti-parallel dimers [145], the scFv domains of the scFv:TRAILR2-p53tet fusion protein presumably protrude to opposing directions. It is thus unclear how this variant works at the molecular level. One possibility could be that enforced clustering of four TRAILR2 molecules, possibly in combination with the weak auto-affinity of TRAILR2, is sufficient to allow the formation of a procaspase-8 filament assembly-inducing TRAILR2-FADD platform. Alternatively, opposing scFv domains could bind to different cells to act reciprocally as anchors to promote secondary clustering of dimeric scFv-TRAILR2 complexes spontaneously by the mechanisms described above for FcγR-bound antibodies.

The overwhelming importance of valency for activation of TRAIL death receptors is also evident from various studies analyzing the agonism of non-conventional TRAIL death receptor binders. For example, using a fibronectin type III scaffold-based TRAILR2 binder and multivariant variants derived thereof, Swers et al. found that binding affinity and valency cumulatively enhance agonistic activity [146]. In particular, they described an octameric TRAILR2 binder with >100 fold higher specific activity than soluble TRAIL in vitro and potent anti-cancer activity in vivo [146,147]. Similarly, it has been found that a TRAILR2-binding peptide elicits strong agonism when fused to a hepatemeric C4b scaffold [148].

### 5.2. Cell Surface Anchored Anti-TRAIL Death Receptor Antibody Variants

As discussed above, cell surface anchored fusion proteins of TRAIL as well as FcγR-bound antibodies display high membrane TRAIL-like activity. It is thus not really surprising that bispecific antibodies and antibody fragments recognizing TRAIL death receptors and a second cell surface-exposed target act as strong agonists upon anchoring to the latter (Figure 7). For example, enhanced agonistic activity of anti-TRAILR2 antibodies have been reported for antibody variants with a heavy chain fused with a N- or C-terminal scFv anchoring domain [75,149]. Vice versa, N-terminal fusion of a TRAILR2-spcecific scFv domain to a MCSP- or FOLR1-specific IgG1 also resulted in increased anchoring-dependent agonism of the scFv-domain [150,151]. Noteworthy, in the case of the anti-MCSP anchored scFv:TRAILR2, there was further enhancement upon FcγR binding of the construct, suggesting that the agonistic potential of the TRAILR2-specific scFv domain was not fully unleashed. Bispecific variants of anti-TRAILR2 antibodies with high anchoring-dependent agonism and good in-vivo efficacy have also been generated by the knob-into-hole technology and a CrossFab unit specific for the tumor stroma antigen FAP [152].

TRAIL death receptor antibody variants with high anchoring-dependent agonism appear attractive two reasons in particular. First, these agonists share with conventional antibodies their excellent pharmacokinetic properties and high stability. Second, similarly to the TRAIL fusion proteins with an anchoring domain, this type of construct principally allows tumor-localized activation of TRAIL death receptors when an anchoring domain is available, recognizing a tumor-specific target. This way, hepatotoxic side effects, as observed with some highly active TRAILR2 agonists, should be reducible.

## 6. Conclusions

A considerable number of clinical trials have been performed with recombinant soluble TRAIL and conventional antibodies with the aim to trigger apoptosis/cell death in tumor cells. These efforts, however, were not convincing yet. There was indeed very good tolerance of these TRAIL death receptor targeting biologicals, but their therapeutic efficacy was poor and disappointing. This outcome is less surprising in view of the evidence in recent years that soluble TRAIL and anti-TRAILR1/2 antibodies typically fail to unleash the full apoptotic activity of the TRAIL death receptors.

The improved understanding of the molecular mechanisms of TRAIL death receptor activation, however, prompted the development of next generation TRAIL- and antibody-based TRAIL death receptor agonists with much higher specific activity reaching those of the highly-active natural TRAILR1/2 stimulator membrane TRAIL. These novel highly-active TRAILR1/2 agonists may now fulfill, with delay, the hopes set in TRAIL death receptor targeting as a therapeutic strategy to treat cancer. There is admittedly evidence that strong systemic TRAIL death receptor activation elicits hepatotoxicity, but this also might not hinder the success of the novel TRAIL death receptor agonists as some of them possess targeting-dependent, and thus, locally-restricted activity.

## Figures and Tables

**Figure 1 cancers-11-00954-f001:**
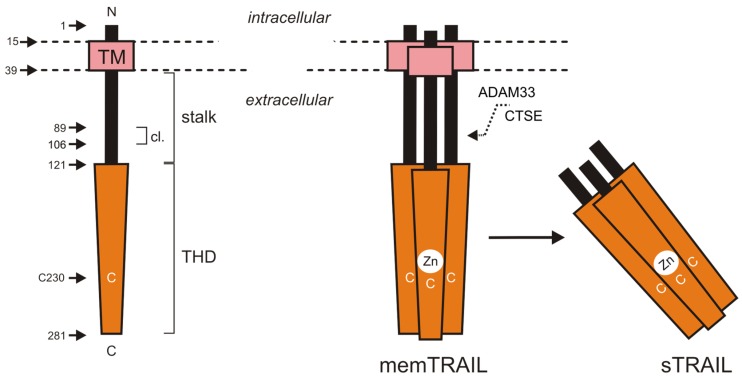
Domain architecture of TRAIL. “cl” indicates a region (aa 89–106) within the stalk region containing one or more sites for proteolytic processing [3]; THD, TNF homology domain; TM, transmembrane domain. For more details see main text.

**Figure 2 cancers-11-00954-f002:**
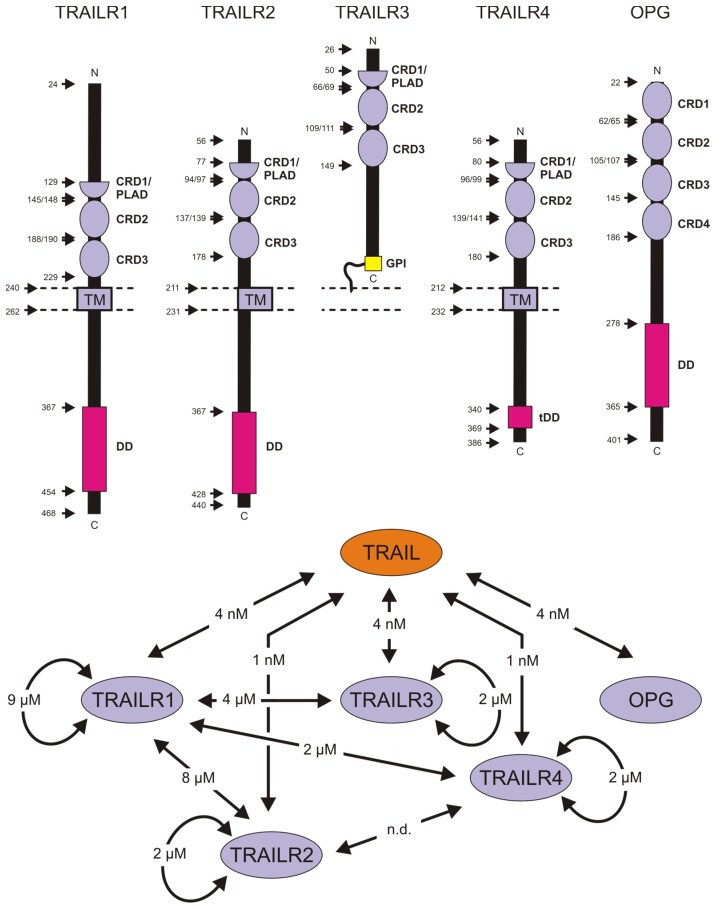
Domain architecture of the TRAIL receptors and their interactions. CRD, cysteine-rich domain; DD, death domain; GPI, glycosylphosphatidylinositol; PLAD, pre-ligand binding assembly domain; TM, transmembrane domain. In contrast to the CRDs, the PLAD is functionally and not structurally defined and matches with CRD1. Please note OPG is a soluble protein; its DD is not involved in the activation of intracellular signaling cascades. Affinities of ligand-independent receptor interactions and for TRAIL binding are indicated [41,42].

**Figure 3 cancers-11-00954-f003:**
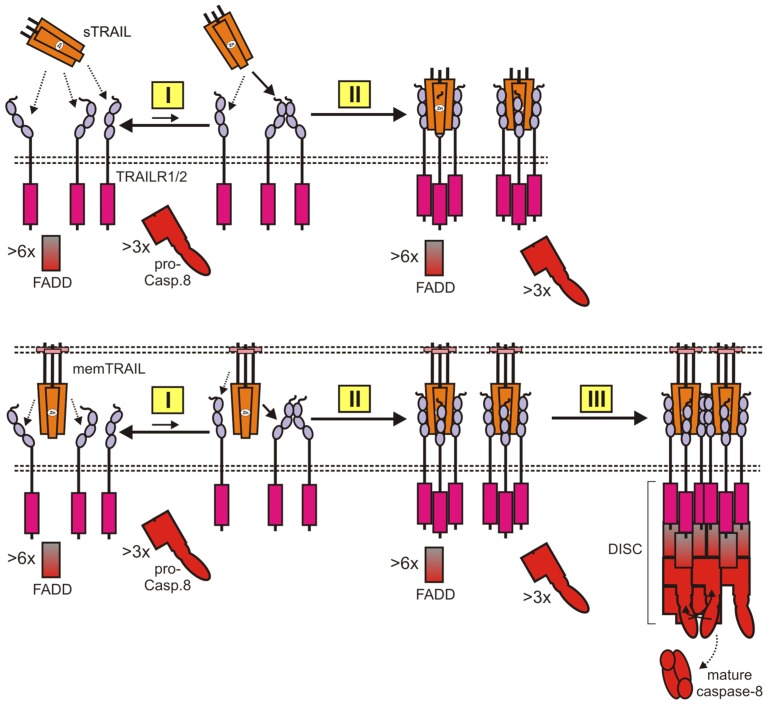
Two-step formation of apoptotic TRAIL death receptor-signaling complexes by TRAIL trimers. In the absence of TRAIL, TRAIL receptors interact with low affinity and thus presumably occur in a dynamic equilibrium of monomeric and dimeric (or trimeric) receptor species (I). Shown is the situation for a single TRAIL death receptor type but please note that TRAIL death receptors and TRAILR3 and 4 also interact in a homo- and heterotypic manner. A TRAIL trimer binds with high-affinity three TRAIL receptor molecules (II). The details of the assembly of this receptor complex from the unliganded receptor species are poorly understood. Evidence for its existence are the crystallographic studies and the fact that crosslinking of TRAIL trimers enhances their apoptotic activity. In the case of membrane TRAIL, trimeric TRAIL-TRAIL receptor complexes spontaneously cluster secondarily to oligomeric networks due to their high local concentration and the weak auto-affinity of TRAILRs (III). The oligomeric receptor network allows six neighboring TRAIL death receptors to form a cap together with FADD that can serve as a condensation nucleus for procaspase-8 filaments (III). In the filaments, two procaspase-8 molecules dimerize and mature by induced proximity and autocatalytic processing.

**Figure 4 cancers-11-00954-f004:**
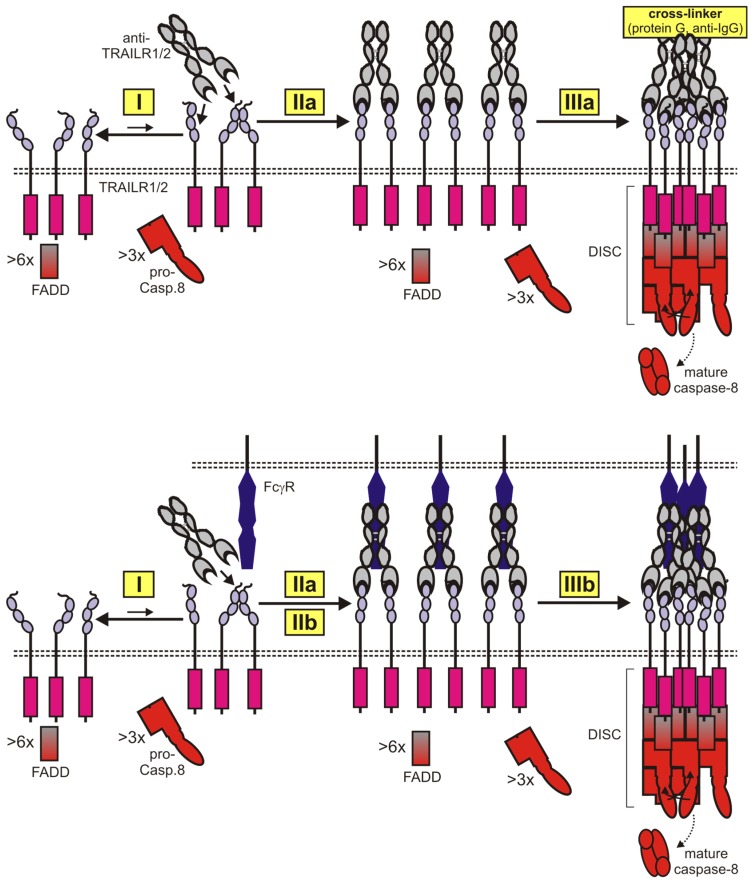
Two-step formation of apoptotic TRAIL death receptor-signaling complexes by bivalent TRAIL death receptor antibodies. In the absence of antibody, TRAIL receptors presumably occur in a dynamic equilibrium of monomeric and dimeric (or trimeric) receptor species (I). A TRAIL death receptor antibody binds with high-affinity two-receptor molecules. This complex is largely inactive and typically does not cluster (IIa). In the presence of FcγR-expressing cells, the antibodies may bind concomitantly to the targeted TRAIL death receptor and FcγRs (IIb). Dimeric antibody-TRAILR1/2 complexes cluster secondarily to active oligomeric networks either due to physical crosslinking of the antibodies (IIIa) or due to the high local concentrations generated by FcγR binding.

**Figure 5 cancers-11-00954-f005:**
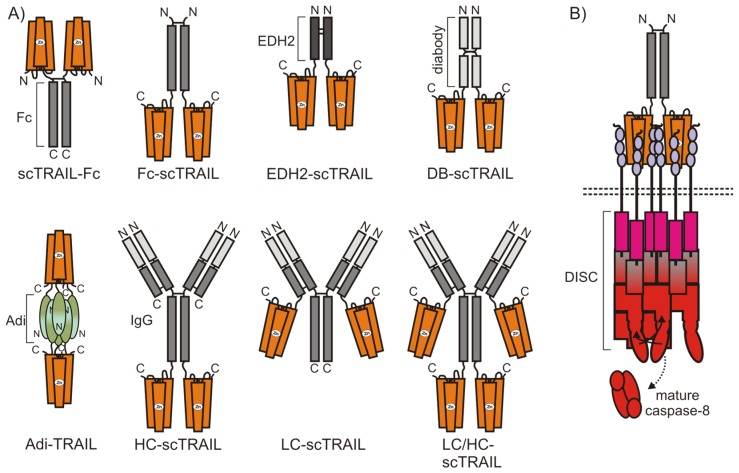
Domain architecture of oligomeric TRAIL fusion proteins (**A**). Apoptotic TRAILR1/2 clustering is illustrated in (**B**) for Fc-scTRAIL.

**Figure 6 cancers-11-00954-f006:**
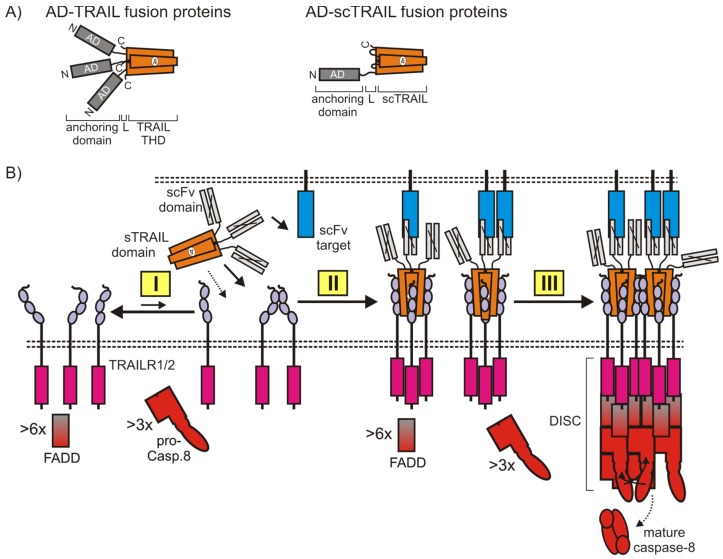
Cell surface anchored TRAIL fusion proteins act similarly to the membrane TRAIL. (**A**) General structure of anchoring domain (AD) TRAIL fusions proteins. Any kind of protein domain which interacts with a cell surface exposed target (scFvs, ligands …) can be considered an anchoring domain. A crucial prerequisite for anchoring-dependent agonism is only that the AD does not result in oligomerization of TRAIL trimers. (**B**) Equilibrium between monomeric and auto-associated TRAIL death receptors (I). scFv-TRAIL interacts with high affinities with TRAIL death receptors and the scFv-recognized cell surface antigen (II). scFv-anchored trimeric TRAIL-TRAILR1/2 complexes secondarily cluster to active oligomeric networks due to the high local concentrations in the cell-to-cell contact zone and the low receptor intrinsic auto-affinity. For more details, see the main text and Figure 3 and Figure 4.

**Figure 7 cancers-11-00954-f007:**
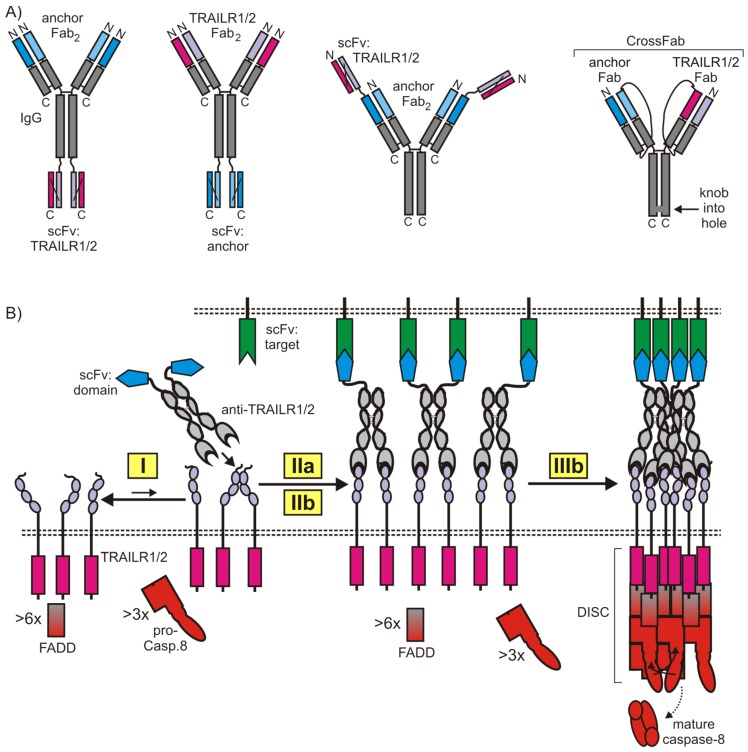
(**A**) Structure of recently published bispecific anti-TRAILR1/2 antibody variants with anchoring-dependent agonism [75,150,151]. (**B**) Mode of action of an anti-TRAILR2 fusion protein with anchoring domain on the heavy chain. For more details, see also the main text and Figure 3, Figure 4 and Figure 6.

**Table 1 cancers-11-00954-t001:** Association of serum levels of TRAIL and disease.

Disease	Soluble TRAIL Level	Correlation with Clinical Relevant Factors	Reference
Hepatitis B virus infection	Increased	Yes, liver damage	[7]
Systemic lupus erythematosus	Increased	No	[8]
Amyotrophic lateral sclerosis	Reduced	No	[9]
Chronic kidney disease	Reduced	Yes, inverse association with mortality risk	[10]
Systemic sclerosis	Increased	Yes, associated with pulmonary involvement	[11]
Recurrent miscarriage	Increased	Not investigated	[12]
Preeclampsia ^1^	Decreased	Not investigated	[13]
Multiple sclerosis	Reduced	No	[14]
Type I diabetes	Reduced	Not investigated	[15]
Hypercholesterolemia	Increase	Yes, low-density lipoprotein	[16]
Polymyositis and Dermatomyocitie	Increased	Yes, disease activity	[17]

^1^ TRAIL measured in maternal plasma.

**Table 2 cancers-11-00954-t002:** TRAILR1- and TRAILR2-specific antibodies ^1^.

Name	Isotype	Increase of Agonistic Activity		Reference
		by Crosslinking ^2^	by FcγR-Binding ^3^	
TRAILR1/DR4				
4H6	mIgG1	>1000		[72]
4G7	mIgG2a	>1000		[72]
DR4-A		>100		[73]
Mapatumumab (HGS-ETR1)	hIgG1	Highly active w/o crosslinking		[74]
DJR1	mIgG1		Considerable	[75]
HS101	mIgG1		Considerable	[75]
TRAILR2/DR5				
CS-1008 (from TRA-8, Tigatuzumab)	hIgG1	10 to >1000		[76]
KMTR2	hIgG1	~3; already highly active w/o crosslinking		[77]
LBY135	hIgG1	>100		[73]
Conatumumab (AMG655)	hIgG1	100 to >1000		[78]
Drozitumab Apomab	hIgG1	>100	Considerable	[79,80]
LexatumumabHGS-ETR2	hIgG1	>10		[81]
Zaptuzumab (AD5.10)	hIgG1	Highly active w/o crosslinking		[82]
DJR2	mIgG1		Considerable	[75]
D-6	mIgG1		Considerable	[75]
Anti-mDR5 (MD5-1)	Hamster IgG		Considerable	[83,84,85]

^1^ If not otherwise stated, antibodies are raised against the human TRAIL death receptors. ^2^ Activity increase is defined as the EC_50_-value of the antibody alone divided by the EC_50_-value of the antibody upon crosslinking (anti-IgG, protein A/G). ^3^ The effect of FcγR binding on agonistic activity has only been tested at one or very few antibody concentrations in most studies. Quantification of the enhancing effect of FcγR-binding was thus not possible.

**Table 3 cancers-11-00954-t003:** Stabilized soluble TRAIL variants.

Name	Stabilization Strategy	Activity Increase ^1^	Oligomerization Effect ^2^	Reference
LZ-TRAIL	N-terminal leucine zipper	3 to >100 (cell-type dependent)	n.d. ^3^	[98]
CPT	135-281-L-121-134	20–60	n.d.	[60,99]
TNC-TRAIL	N-terminal tenascin-C trimerization domain	5–10	250–1000	[2]
ST	Coiled-coil of SP-D	~10	n.d.	[96]
scTRAIL	Peptide linker connected TRAIL protomers	scTRAIL has only been published as part of scTRAIL fusion proteins		[97]
HA5FT	Ad5 fiber	3–5	n.d.	[100]
HA5ST	Ad5 shaft	3–5		
sfTRAIL	N-terminal foldon domain ^4^	n.d.	n.d.	[101]

^1^ Activity increase is defined as the EC50-value of conventional soluble TRAIL divided by the EC50-value of the stabilized TRAIL variant listed in the table. The differences to conventional soluble TRAIL are often more pronounced in vivo. ^2^ Oligomerization effect is defined as the ratio of the EC50-values of the stabilized TRAIL variant without and with oligomerization. ^3^ not determined. ^4^ Trimerization foldon domain from the fibritin protein of the bacteriophage T4.

**Table 4 cancers-11-00954-t004:** Fusion proteins with two or more “TRAIL trimer” domains.

Name	Oligomerization Strategy	Number of TRAIL Domains ^1^	Activity Increase ^2^	Reference
APG350 ^3^	C-terminal IgG1 Fc domain	2	>100	[105]
Fc-scTRAIL	N-terminal IgG1 Fc domain	2	10	[108]
Db-scTRAIL		2	10–30	[108]
EDH2-scTRAIL	N-terminal heavy chain domain 2 of IgE	2	10	[108,109]
LC-scTRAIL	Fused to light chain of IgG1	2	n.d. ^4^	[110]
HC-scTRAIL	Fused to heavy chain of IgG1	2	n.d. ^4^	[110]
LC/HC-scTRAIL	Fused to light and heavy chain of IgG1	4	n.d. ^4^	[110]
Adi-TRAIL	Fusion to arginine deiminase	2	n.d. ^4^	[111]

^1^ TRAIL domain means three covalently or non-covalently assembled TRAIL protomers. For domain architecture, see Figure 4. ^2^ Activity increase is defined as the EC50-value of conventional soluble TRAIL divided by the EC50-value of the oligomeric TRAIL fusion protein. ^3^ Also designated as vhTRA. ^4^ not determined, thus no dose response comparison with conventional trimeric TRAIL were shown in publications but data shown indicate high apoptotic activity of the TRAIL variants.

**Table 5 cancers-11-00954-t005:** Soluble TRAIL fusion proteins with cell anchoring-restricted activity. Fusion proteins, for which it is unclear whether their enhanced activity is due to cell anchoring or oligomerization, are not listed (e.g., IL2-TRAIL, ref. [111]; TMPT1-sTRAIL, Ref. [112]).

Anchoring Domain	Anchor Target	Activity Increase ^1^	Effect of Anchoring Domain	Reference
scFv:FAP	FAP	~20		[3]
scFv:C54	EpCAM (EGP2)	>100		[113]
scFv:425	EGFR	- ^2^		[114]
scFv:CD7	CD7	>100		[115]
scFv:425	EGFR	~50 ^3^		[116]
scFv:CD19	CD19	- ^2^		[117]
scFv:CD33	CD33	>50		[118]
scFv:MCSP	MCSP	>100		[119]
scFv:ErbB2	ErbB2 ^4^	~10		[97]
K12	CD7 ^5^	>>100		[120]
svFv:CD3	CD3 ^5^	>>100		[120]
scFv:62	Kv10.1			[121]
scFv:hu225	EGFR ^4^	~10		[122]
scFv:G28	CD40	~100	Activates CD40 and thus stimulates DC maturation.	[123]
scFv:CD20	CD20	- ^2^		[124]
scFv:Px44	DSG			[125]
scFv:CD70	CD70 ^6^	10–100	Inhibition of CD70–CD27 interaction.	[126]
scFv:CD47	CD47	>50	Blocks CD47-SIRPα interaction and abrogates inhibition phagocytosis.	[127]
scFv:M58	MRP-3			[106,128]
scFv:hu225-EHD	EGFR	10–20		[106]
scFv:CLL-1	CLL1 ^5^	>>100		[129]
scFv:PD-L1	PD1	>100	Blocks PD1–PD–L1 interaction.	[130]
scFv-EHD	EGFR	3–5		[108]
CD40ed	memCD40L	>100	Blocks antiapoptotic CD40L signaling.	[131]
RGD	αVβ3, αVβ5	- ^2^		[132]
Mesothelin	Muc16	>10		[133]
Meso(1-64)	Muc16	>10		[134]
CD19L	CD19	- ^2^		[135]
ENb	EGFR	- ^2^	Blocks EGFR signaling.	[136]
Z	PDGFRß	~4		[137]

^1^ Activity increase is defined as the EC50-value of the non-anchored TRAIL fusion protein divided by the EC50-value of the anchored molecule. ^2^ No dose response data are given, but strong reduction of apoptotic activity upon blocking access to the anchor target was shown. ^3^ A mutated TRAIL domain with reduced TRAILR2 binding has been used in this study. ^4^ scTRAIL domain was used as a soluble TRAIL domain. ^5^ Cell anchoring of soluble TRAIL was used here to arm largely TRAIL-resistant T-cells and granulocytes with additional cytotoxic activity. ^6^ Effects were described with wt TRAIL domain but also with TRAIL domains with reduced binding of TRAILR1 or TRAIL2.

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
