# Peer review of "Molecular Mode of Action of TRAIL Receptor Agonists—Common Principles and Their Translational Exploitation"

_cancers, 2019, doi:10.3390/cancers11070954_

Round 1

Reviewer 1 Report

This is a comprehensive review of TRAIL receptor agonists. The only concern I see is the presence of numerous spelling grammatical errors that a critical read would correct.

Author Response

We have revised the manuscript regarding grammar and spelling. We also split over-long and complicated sentences. Please see also responses to other reviewers.

Reviewer 2 Report

The review written by Prof. Wajant is nicely and clearly presented and add value and interpretations to the field. Below are few suggestions to improve the text and increase impact:

- Language editing few sentences in the beginning of the introduction and first two sections I had to read several times to understand the content due to formulation.

- Which diseases and what are the consequences of TRAIL Row59

- Short about why nonimmune cells express TRAIL mentioned intestinal and mammary epithelial cells Row 67-68.

- A little confusing whether TRAIL is dependent or independent of p53 as it is controversially mentioned in row 38 and 268. Please clarify.

- Start with introductory statement in the beginning av new sections like you do in 5 (row 417).

- Very little is mentioned about the immune cells expressing TRAIL it would be of high interest for immunologists to read about the TRAIL signaling beyond apoptosis e.g. migration.

Author Response

- Language editing few sentences in the beginning of the introduction and first two sections I had to read several times to understand the content due to formulation.

We have revised the manuscript regarding grammar and spelling. We also split over-long and complicated sentences.

- Which diseases and what are the consequences of TRAIL Row59

We included a table (Table 1) listing the most relevant diseases associated with increased or lowered serum TRAIL levels. These studies only report correlations. Thus, it is unclear whether TRAIL is of functional relevance in these diseases. This is briefly addressed in the main text. We also referred to the evidence for TRAIL being a tumor surveillance factor (l. 69-73).

- Short about why nonimmune cells express TRAIL mentioned intestinal and mammary epithelial cells Row 67-68.

The biological relevance of TRAIL expression by these cell types is largely unknown. We stated this in the revised manuscript.

- A little confusing whether TRAIL is dependent or independent of p53 as it is controversially mentioned in row 38 and 268. Please clarify.

It was meant that the TRAIL-induced apoptosis pathway per se does not include p53 activation. However, TRAILR2 expression can be increased by p53. We therefore skipped the confusing notion that TRAIL-induced apoptosis is p53-indpendent as it is not of relevance for the topic of the review.

- Start with introductory statement in the beginning av new sections like you do in 5 (row 417).

We think an introductory statement in section 5 is helpful to highlight the similarities to section 4 but this a special situation and in the other sections we do not feel that an introductory statement is helpful.

- Very little is mentioned about the immune cells expressing TRAIL it would be of high interest for immunologists to read about the TRAIL signaling beyond apoptosis e.g. migration.

We mentioned the role of immune cell-expressed TRAIL in tumor surveillance. With respect to non-apoptotic TRAIL signaling, we referred to other reviews that cover this issue in detail.

Reviewer 3 Report

The manuscript entitled ‘Molecular mode of action of TRAIL receptor agonists - common principles and translational pitfalls’ by Harald Wajant comprehensively demonstrated about TRAIL receptor and agonists. This area is quite interesting research topic and the manuscript is well organized. Please check below minor comments to improve the quality.

1. Line 7: Correspondence:orrespondence

2. Line 23: und à and

3. Line 61: Need to add the abbreviation of cathepsin E

4. For the upper panel in Figure 2. indicate which is TRAILR1, TRAILR2 etc….. for the reader

5. It needs to draw a line between TRAILR2 and TRAILR4 to indicate Kd> 1 μM in Fig. 2?

6. Line 80-81: a truncated seemingly non-functional DD à add (tDD)

7. Line 83-83: OPG (osteoprotegerin) à switch the order

8. Line 89: “GPI, glycosylphosphatidylinositol; PLAD, pre-ligand binding assembly domain” à they are not indicated in Figure.

9. Line 199, 211, 221, 227, 230, 232, 234: check the typo ‘be’, ‘it’, ‘lesions’, ‘form’, ‘Dularnim’, ‘inorelbine’, ‘therpeutically’, ‘oligoemerization’, ‘it’, ‘permutet’ and please take this opportunity to check any other typos throughout the manuscript

10. Line 239: ‘several groups and companies developed agonistic antibodies ~~ A couple of them have been extensively~’ à It looks like that dulanermin is introduced as an antibody drug.

11. In table 1, the meaning of ‘10 - > 1000’ is not clear. In table 2, for example, ‘3 to > 100’ or ‘5 – 10’ looks better. Also, check the reference #72, 73

12. Line 260: KMTR2?

13. For the paragraph for 5.2 (from line 464), add (Figure 7) to an appropriate area.

14. No (A) and (B) in Fig.7. Only in legend.

15. Lines 269-281 or 489: It looks like that brief description about the drug delivery system can be helpful for the targeting.

Author Response

1. Line 7: Correspondence:orrespondence --- corrected

2. Line 23: und à and  --- corrected

3. Line 61: Need to add the abbreviation of cathepsin E  --- included

4. For the upper panel in Figure 2. indicate which is TRAILR1, TRAILR2 etc….. for the reader   --- corrected

5. It needs to draw a line between TRAILR2 and TRAILR4 to indicate Kd> 1 μM in Fig. 2? --- line has been included w/o concrete affinity value due to lack of available quantitative data for this interaction in the literature.

6. Line 80-81: a truncated seemingly non-functional DD à add (tDD)   --- corrected

7. Line 83-83: OPG (osteoprotegerin) à switch the order  --- corrected

8. Line 89: “GPI, glycosylphosphatidylinositol; PLAD, pre-ligand binding assembly domain” à they are not indicated in Figure. --- now included in the figure

9. Line 199, 211, 221, 227, 230, 232, 234: check the typo ‘be’, ‘it’, ‘lesions’, ‘form’, ‘Dularnim’, ‘inorelbine’, ‘therpeutically’, ‘oligoemerization’, ‘it’, ‘permutet’ and please take this opportunity to check any other typos throughout the manuscript  --- corrected/done

10. Line 239: ‘several groups and companies developed agonistic antibodies ~~ A couple of them have been extensively~’ à It looks like that dulanermin is introduced as an antibody drug. ---   The misleading sentence in line 239 was rephrased (l. 243/244).

11. In table 1, the meaning of ‘10 - > 1000’ is not clear. In table 2, for example, ‘3 to > 100’ or ‘5 – 10’ looks better. Also, check the reference #72, 73

In table 1 (now table 2) “10 - > 1000” etc. has been replaced by “10 to > 1000” for clarity. References are checked. Indeed, the second Li reference was wrong and has been replaced by the correct Li reference. Please note, there are still 2 papers by Li and Ravetch citedin this context.

12. Line 260: KMTR2?  ---   corrected

13. For the paragraph for 5.2 (from line 464), add (Figure 7) to an appropriate area.  ---   done

14. No (A) and (B) in Fig.7. Only in legend.   ---   now included in the figure, too.

15. Lines 269-281 or 489: It looks like that brief description about the drug delivery system can be helpful for the targeting. --- We aplogize for not including a detailed and sophisticated discussion of drug delivery systems for TRAIL agonists (e.g. liposomes, nanobodies, viral vectors etc.) since we feel that this distract the reader from the central topic of the review "molecular mode of action of agonists".

Reviewer 4 Report

In this manuscript, entitled " Molecular mode of action of TRAIL receptor agonists: common principles and common pitfalls” tried to summarize the need of TRAIL death receptor agonists and molecular mechanism of TRAIL-induced cell death signaling.

The comments for this manuscript are as follows-

1.       In line 23, und should be replaced with and.

2.       The author should include some more recent references:

Oncotarget. 2018 Feb 17;9(21):15566-15578. doi: 10.18632/oncotarget.24526. eCollection 2018 Mar 20.Relationship between the agonist activity of synthetic ligands of TRAIL-R2 and their cell surface binding modes.

Oncotarget. 2017 Jul 4;8(27):44232-44241. doi: 10.18632/oncotarget.17790.Synergistic targeting of malignant pleural mesothelioma cells by MDM2 inhibitors and TRAIL agonists.

Tumour Biol. 2017 May;39(5):1010428317699120. doi: 10.1177/1010428317699120. Focal adhesion kinase inhibitor PF573228 and death receptor 5 agonist lexatumumab synergistically induce apoptosis in pancreatic carcinoma.

Cell Death Differ. 2017 Mar;24(3):500-510. doi: 10.1038/cdd.2016.150. Epub 2017 Feb 10. N-glycosylation of mouse TRAIL-R and human TRAIL-R1 enhances TRAIL-induced death.

Invest New Drugs. 2017 Jun;35(3):298-306. doi: 10.1007/s10637-016-0420-1. Epub 2017 Jan 3. First-in-human study of the antibody DR5 agonist DS-8273a in patients with advanced solid tumors.

Front Oncol. 2015 Apr 2;5:69. doi: 10.3389/fonc.2015.00069. eCollection 2015. Trailing TRAIL Resistance: Novel Targets for TRAIL Sensitization in Cancer Cells.

Expert Rev Precis Med Drug Dev. 2018;3(3):197-204. doi: 10.1080/23808993.2018.1476062. Epub 2018 May 28. TRAIL pathway targeting therapeutics.

3. The manuscript is well written with a clear objective.

Author Response

I'am a bit confused. I received an Email from Sara Radunovic June 24th containing comments of a delayed 4th reviewer. These comments differ from those available here from the platform for reviewer 4. I therefore addressed here the comments received by Email plus the comments listed here on the platform. 

EMAIL 24th

Dear Dr. Wajant,

May I kindly ask to address the following comments of the reviewer about your manuscript, accordingly:

"I have read the manuscript by Dr Harald Wajant entitled as  “Molecular mode of action of TRAIL receptor agonists 2 – common principles and translational pitfalls”.
In general, the manuscript appears to comprehensively cover the relevant literatures with careful interpretation and analysis. The main stream of text is logical and comprehensive, it seems. However, I would suggest author to improve the quality of individual figures and tables to get the point. I feel that these current figures may not easy to capture the point at glance in particular how these different structure of TRAIL are connected to the different antibodies and hence resulted in the pit falls of the therapy using these antibodies.

We agree the title “Molecular mode of action of TRAIL receptor agonists – common principles and translational pitfalls„ is somewhat misleading with respect to “…pitfalls”. We have thus changed the title to “Molecular mode of action of TRAIL receptor agonists – common principles and their translational exploitation“

The figures, in general, are not get the points at a glance as to what authors intent to illustrate. Figures (Fig 3, Fig4, Fig 6 and Fig 7) appear to be similar each other, and somewhat redundant, how these figures can connect the pit fall of molecular mode of action of TRAIL receptor agonists 2 – common principles and translational pitfalls”. I would encourage author to improve the presentation of the individual figures more specific focus, as current figures appear to be redundant.

The redundancy in figures 3 and 4 reflects the related mode of action of antibodies and soluble TRAIL, thus the relevance of oligomerization for their agonism. The redundancy of figures 6B and 7B with the membrane TRAIL panel of figure 3 (lower part) is due to the fact that cell surface-anchored TRAILR1/2-antibodies/TRAIL act similar to membrane TRAIL.  Due to the overwhelming relevance of oligomerization and cell attachment for the activity of TRAIL DR-addressing molecules, we think this redundancy is acceptable and underscores the generic relevance of these factors for the development of TRAIL DR agonists.

Partly due to insufficient legend, Tables (all of three tables) are not easy to understand the differences of the antibodies. For example, Table 1, the column, “strongly enhancing” is not clear what this specifically and scientifically mean to these individual antibodies listed on the left column.

We replaced “strongly enhancing” in the “Activity increase by FcR-binding” column by “considerable”. Unfortunately, the relevance of FcgR-binding of anti-TRAIL-DR antibodies has typically only be demonstrated “qualitatively” in the various publications for one antibody concentration. It was thus not possibly to give hard numbers as in the “Activity increase by oligomerization” column. We indicated this now in a footnote of the table.

Table 2. it is not clear “ how does “activity increase mean” scientifically.

“Activity increase” is defined in the footnote 1 of the table: “Activity increase is defined as the EC50-value of apoptosis induction in vitro of conventional soluble TRAIL divided by the EC50-value of the stabilized TRAIL variant listed in the table.” The values are based on “apoptosis induction” experiments with conventional TRAIL and the TRAIL variants listed in the table which were performed in the cited studies. In the revised version, we included “of apoptosis induction in vitro” in the definition for better understanding. 

Table3. It is not clear the title as “ Fusion proteins of soluble TRAIL with multiple TRAIL means? “ What does authors mean by “multiple”?

Table 3: We rephrased the table title to “Fusion proteins with multiple “TRAIL trimer” domains.” We further include in a footnote that “TRAIL domain means three covalently or non-covalently assembled TRAIL protomers”. We replaced “multiple” by “2 or more”. Last but not least we referred in the footnote of the table to figure 4 showing schemes of the molecules listed in table 3.

I also would suggest author to some more details of the antibody affinity in Table 4.

We agree that a column with the affinities of the scFv/anchoring domain fusion proteins for the cell surface expressed target structures would be interesting. Unfortunately, with exception of 2 studies, these information is not given in the cited studies. It would been possible to search in secondary literature for the affinities of the scFvs. However, this would not be productive as the affinities of the monomeric scFvs gives no solid information about the affinity of the scFvs as trimers what is the relevant issue for the TRAIL fusion proteins.

How these different antibodies are connected individually to the different forms of the TRAIL illustrated in the Figures may not be clear.

In the new figure 6A, we show how the anchoring domains (typically scFvs) are linked to soluble TRAIL (or in some cases scTRAIL).

There have been series of recent reviews published, which discussing somewhat similar subjects of weakness and complexities of modulation of the TRAIL signal for cancer therapy[1-4]. The current manuscript is somewhat redundant, but more focusing the antibodies of TRAIL signal which would be a good compensation of other relevant review for TRAIL mediated therapy on cancer. However, I would encourage authors to refer these literatures more precisely of the topic and possibly discuss some extent in particular #1 and 4 and one of the topic on ONC201 which is referred and discussed by Yuan in Ref #2 [1-4].

Suggested ref. [1] has already been cited as ref.[ 59] (revised version . We cited this review which gives an excellent view over the whole “TRAIL” field now more frequently, especially when issues are mentioned which are not in the focus of our manuscript. In a similar way we used/cited suggested ref. 2. We not included suggested ref.3 and 4. since all issues addressed in this reviews from 2008 and 2012  are also covered by the suggested reviews 1 and 2 and the already cited review by Siegmund et al  which are from 2017 and 2018.

We briefly mentioned Onc201 briefly as a potential therapeutic activating  (at least in part via the TRAIL/TRAILR system as suggested (new ref. x an y). We abstained, however, from discussing it in detail as it acts indirectly and not via direct TRAIL DR activation.  

I feel that reference 4 (Table 1) precisely and comprehensively refer individual problems of antibody-based therapies of TRAIL and relevant antibodies.

The below may need attention.
Page 7 #. Line 6” This is maybe not really---” may better to improve the grammar. Perhaps it would be better to state “This may not be---"

Phrase “This is maybe not really …..” has been improved to “This is less surprising when one considers that Dulanermin…..”

1.       In line 23, und should be replaced with and. ---   corrected

2.       The author should include some more recent references:

Oncotarget. 2018 Feb 17;9(21):15566-15578. doi: 10.18632/oncotarget.24526. eCollection 2018 Mar 20.Relationship between the agonist activity of synthetic ligands of TRAIL-R2 and their cell surface binding modes.

Oncotarget. 2017 Jul 4;8(27):44232-44241. doi: 10.18632/oncotarget.17790.Synergistic targeting of malignant pleural mesothelioma cells by MDM2 inhibitors and TRAIL agonists.

Tumour Biol. 2017 May;39(5):1010428317699120. doi: 10.1177/1010428317699120. Focal  adhesion kinase inhibitor PF573228 and death receptor  5 agonist lexatumumab synergistically induce apoptosis in pancreatic  carcinoma.

Cell Death Differ. 2017 Mar;24(3):500-510. doi: 10.1038/cdd.2016.150. Epub 2017 Feb 10. N-glycosylation of mouse TRAIL-R and human TRAIL-R1 enhances TRAIL-induced death.

Invest New Drugs. 2017 Jun;35(3):298-306. doi: 10.1007/s10637-016-0420-1. Epub 2017 Jan 3. First-in-human study of the antibody DR5 agonist DS-8273a in patients with advanced solid tumors.

Front Oncol. 2015 Apr 2;5:69. doi: 10.3389/fonc.2015.00069. eCollection 2015. Trailing TRAIL Resistance: Novel Targets for TRAIL Sensitization in Cancer Cells.

Expert Rev Precis Med Drug Dev. 2018;3(3):197-204. doi: 10.1080/23808993.2018.1476062. Epub 2018 May 28. TRAIL pathway targeting therapeutics.

We included  the CDD paper. The two reviewes were not included because to the suggestion of the other reviewer 4 ("email comments" above), we already included two reviewes covering the same isssues (refs. 31 and 37 also from 2017 and 2018). The Oncotarget 2017 ref. uses rec. TRAIL from Amgen, which is already cited in the review, and gives no new functional inside in the mode of action of TRAIL. We think therefore it is not necessary to include this reference. Similarly, lexatumumab is also already covered in the review and the Tumor Biol article does not contribute to the understanding how lexatumumab activates its target.

3. The manuscript is well written with a clear objective.

Many thanks for this appreciation.